# Surface diffusion-limited lifetime of silver and copper nanofilaments in resistive switching devices

Wei Wang [1], Ming Wang[2], Elia Ambrosi [1], Alessandro Bricalli[1], Mario Laudato[1], Zhong Sun [1], Xiaodong Chen [2] & Daniele Ielmini [1]

Silver/copper-filament-based resistive switching memory relies on the formation and disruption of a metallic conductive filament (CF) with relatively large surface-to-volume ratio. The nanoscale CF can spontaneously break after formation, with a lifetime ranging from few microseconds to several months, or even years. Controlling and predicting the CF lifetime enables device engineering for a wide range of applications, such as non-volatile memory for data storage, tunable short/long term memory for synaptic neuromorphic computing, and fast selection devices for crosspoint arrays. However, conflictive explanations for the CF retention process are being proposed. Here we show that the CF lifetime can be described by a universal surface-limited self-diffusion mechanism of disruption of the metallic CF. The surface diffusion process provides a new perspective of ion transport mechanism at the nanoscale, explaining the broad range of reported lifetimes, and paving the way for material engineering of resistive switching device for memory and computing applications.

---

[1] Dipartimento di Elettronica, Informazione e Bioingegneria, Politecnico di Milano and IUNET, Piazza L. da Vinci, 32-20133 Milano, Italy. [2] Innovative Centre for Flexible Devices (iFLEX), School of Materials Science and Engineering, Nanyang Technological University, 50 Nanyang Avenue, Singapore 639798, Singapore. Correspondence and requests for materials should be addressed to X.C. (email: chenxd@ntu.edu.sg) or to D.I. (email: daniele.ielmini@polimi.it)

Resistive switching (RS) memory, technically referred as resistive random-access memory (RRAM), is a two-terminal device that can change its resistance in response to the electrical stimulus by a voltage pulse, as a result of the formation and disruption of a nanoscale conductive filament (CF) with relatively high conductance due to the high concentration of defects. Defects can originate from the host material, such as oxygen vacancies in a metal oxide based memory[1], or from the electrodes via ionic migration across the insulating material, e.g., copper (Cu) or silver (Ag) in the so-called conductive bridge random-access memory (CBRAM)[2]. Depending on the specific application, the formed CF has a lifetime, or retention time, ranging from few microseconds to several months, or even years. For example, RS device finds application in non-volatile memories, where the formed CF must have a lifetime in the range of 10 years[1,3]. RS device also finds application as a volatile switch which is characterized by a short lifetime in the range of microseconds to milliseconds, providing a feasible technology for fast selection devices in crosspoint arrays of memory or sensor devices[4,5]. The tunable lifetime of the CF in the volatile RS device also mimics the short term plasticity of biological synapses[6,7], enabling a burst of novel applications for brain-inspired neuromorphic computing[8–11]. Recently, the coexistence of volatile and non-volatile RS within the same device depending on the compliance current during the CF formation has been extensively reported[12–14]. However, conflicting explanations are being proposed for the CF lifetimes in non-volatile and volatile RS phenomena. In non-volatile switching, the rupture of metallic filament is usually attributed to the out-diffusion of the metallic atoms from the CF to its host dielectric material[15,16]. On the other hand, lifetime in volatile switching is interpreted as the consequence of the atomic clustering to minimize the CF-dielectric interfacial energy[17–19]. A comprehensive physical understanding of CF lifetime in volatile and non-volatile RS device may enable a stronger ability to engineer the device materials and operation toward application-based optimization of RS device.

Here, we show that metallic CF lifetime is strongly affected by surface atomic self-diffusion. The gradient of surface atomic vacancy concentration induces the migration of atoms on the CF surface toward the minimization of the surface area, leading to clustering of metal atoms and eventually to CF disruption. Theoretical analysis predicts that the surface diffusion effect is only dominant in the sub-10 nm scale at room temperature, due to the high surface-to-volume ratio. The lifetime for disruption of a typical CF can span from microseconds to years for the CF size (diameter) changing from <1 nm size to 14 nm size. The size-dependent lifetime is experimentally validated with a broad range of data for various types of both non-volatile and volatile RS devices. This work provides a new perspective of ion transport mechanism in nanoscale, paving the way for structural and material engineering of nanoionic devices.

## Results

### Volatile and non-volatile switching.
Figure 1a shows a RS device based on a metal-insulator-metal (MIM) structure with Au electrodes and a dielectric layer of silk-Ag nanowires (Ag NWs) composite. Ag NWs are buried in the silk layer[20], as shown by the scanning electron microscope (SEM) sectional and planar views in Fig. 1b. Ag NWs serve as reservoirs for metallic atoms for the formation of a nanoscale CF which electrically connects the two electrodes under the application of an external voltage (Fig. 1c). To erase the memory state, the CF can be ruptured by a second external stimulus. These processes are shown by the electrical current-voltage (I–V) characteristics in Fig. 1d, indicating a set transition, or CF formation, at positive voltage, followed by a reset transition, or CF disconnection, at negative voltage. The formed CF usually shows a non-volatile behavior, namely it remains stable after set transition if no further voltage excitation is applied, which forms the basis for the memory operation. Set operation at negative voltage and reset operation at positive voltage can also take place, thanks to the symmetric structure of the nanocontact device where Ag is present in both electrodes (Supplementary Fig. 1 and Supplementary Note 1). The lifetime of the CF should be longer than ten years for a practical memory application where data must be stored reliably. However, volatile switching can also occur, where the CF spontaneously ruptures after formation with a lifetime as short as few microseconds. Volatile and non-volatile switching can coexist within the same device structure, with the transition between the two switching modes being controlled by the compliance current $I_C$, namely the maximum current allowed to flow across the device during the set transition[21–23]. Figure 1e shows typical $I$-$V$ curves for the same device in Fig. 1d operated at lower $I_C$: the RS device shows non-volatile switching for $I_C = 10$ mA, while a lower $I_C$ leads to volatile switching where the CF disconnects spontaneously as the voltage is reduced to zero after the set transition. Bidirectional volatile switching, where the set transition can take place at either positive or negative voltage, is usually demonstrated[23]. The value of the compliance current $I_C$ controls the device behavior, where switching become increasingly volatile as $I_C$ is decreased (Fig. 1f). Since volatile and nonvolatile switching solely differ by the lifetime of their CFs, it is reasonable to attribute the two switching modes to the same mechanism. Since a higher $I_C$ generally corresponds to a thicker CF[1,24], it is also plausible to explain the volatile and non-volatile behaviors to a strong size dependence of CF lifetime. In fact, the nanocontact effect[20] between Ag NWs (Fig. 1c) in our demonstrated device enable a fine control of the filament size by $I_C$ since the limited Ag atoms source and confined electric field eliminate a common overshoot issue during the operation of RS devices[25].

Given the random nature of the Ag NW network in the host material, the distance between the two electrodes in the switching region might be affected by variability. Each device might in fact have a different distance between the active NWs, thus resulting in a variation of threshold voltage from device to device. Even within the same device, the weakest point where the filament is formed might change location or distance at each cycle, thus resulting in a cycle-to-cycle variation of threshold voltage. The threshold voltage distribution of our devices were shown to have a relatively good uniformity both from cycle to cycle, and from device to device[20], thus indicating a relatively small variation of the distance between Ag NWs.

### Surface-diffusion effects.
Recent in situ transmission electron microscope (TEM) observations[17,26] of metallic CFs revealed that Ag and Cu atoms from the CF tend to form clusters rather than out-diffuse toward the host dielectric material. The clustering of the metal atoms leads to the formation of nanoclusters and nanospheres[27]. To highlight the clustering-induced CF rupture, we run a molecular dynamics (MD) simulation by using the LAMMPS program[28] adopting an embedded-atom-method (EAM) potential[29] to describe the interactions between Ag atoms. We set the system temperature to 800 K to accelerate the CF morphological change to a reasonable simulation time. Figure 2a–c shows the time evolution of the simulated Ag CF between the two electrodes, which might represent two Ag NWs in the structure of Fig. 1c, or two planar Ag electrodes in a vertical MIM structure[4,23].

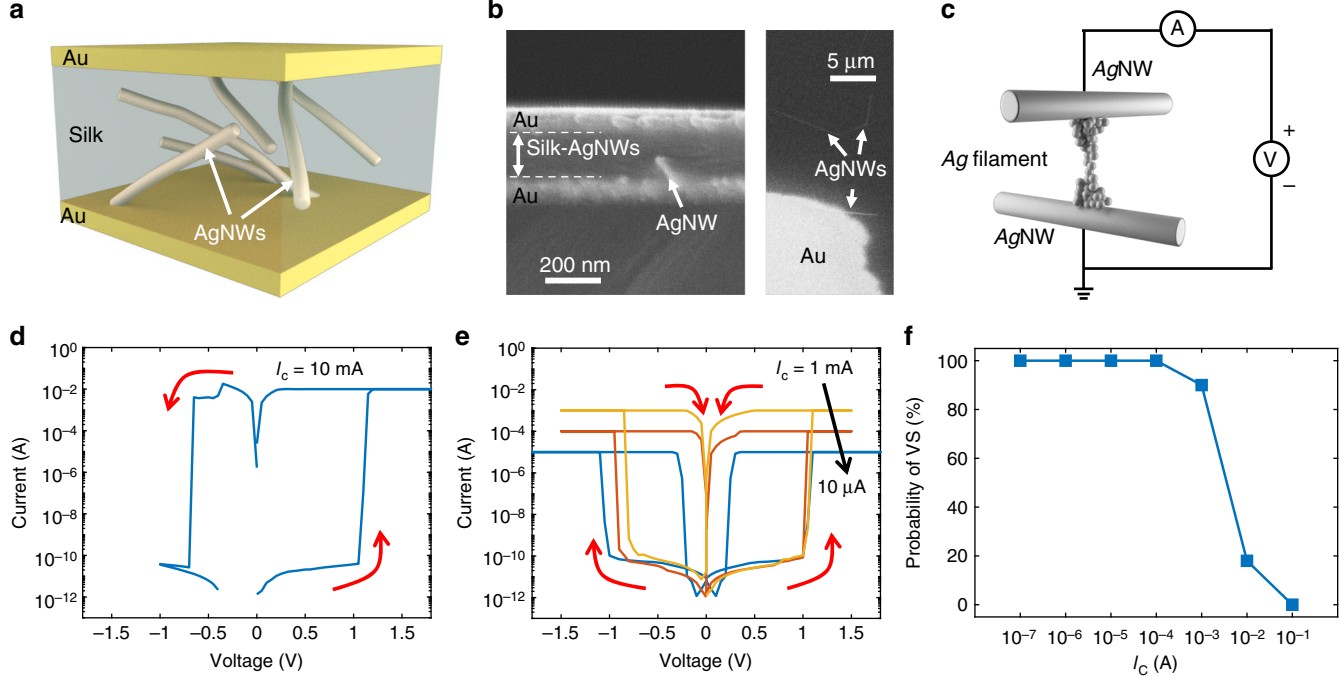

**Fig. 1** Volatile and non-volatile switching of RS devices. **a** Schematic structure of a RS device based on Ag NWs. Bottom and top Au layers act as contact pads, Ag NWs act as effective bottom and top electrodes for the CF formation, and silk acts as insulating switching layer. **b** Sectional (left) and top planar (right) SEM views of the RS device. **c** Illustration of the metallic CF formation between two Ag NWs. A metallic CF can be formed in response to the application of a large voltage to the electrodes. In non-volatile switching, the CF can remain in the connected state for a relatively long time until another voltage pulse induces its disconnection. On the other hand, in the case of volatile switching, the CF retracts back to the electrodes immediately after set transition. **d** I–V characteristics for non-volatile switching for a high compliance current $I_C = 10$ mA. When a positive sweep voltage is applied, the current increases sharply, indicating the formation of the metallic CF. The compliance current $I_C$ across the device controls the size of the formed CF. After the positive voltage sweeps back to zero, the device still remains in the low resistance state. As a negative voltage is applied, the device switches to the high resistance state indicating CF disruption. **e** I–V characteristics for volatile switching of the device at low $I_C$. When a positive sweep voltage is applied, the current increases sharply indicating CF formation. However, the device quickly switches back to its original high resistance state after the forcing voltage is swept to zero. Bidirectional volatile switching is observed, as the application of a negative voltage also generates a metastable CF. **f** Probability for volatile switching (VS) as a function of $I_C$, indicating the transition from volatile switching to non-volatile switching at increasing $I_C$

According to MD simulations, the metallic CF can spontaneously break as a result of atomic surface diffusion driven by the minimization of the system energy. The atomic surface diffusion originates from the gradient of surface atomic vacancy concentration or the gradient of the surface atomic chemical potential[30], resulting in a tendency to minimize the surface energy[31]. Atoms in the bulk of the CF instead remain fixed in their lattice sites of the crystal (Fig. 2d, e, and see Supplementary Fig. 2–3 and Supplementary Note 2 for more details), as confirmed by recent observations by in situ TEM[21,27]. Based on the Gibbs-Thomson effect, the surface atomic flux $J_s$ along an arbitrary surface can be modeled by assuming isotropic surface diffusion, leading to[32]:

$$J_s = -\left(\frac{D_s \gamma \delta^4}{kT}\right) \nabla_s \kappa, \qquad (1)$$

where $D_s$ is the surface diffusion coefficient, $\gamma$ is the surface energy, $\delta$ is the interatomic distance, $k$ is Boltzmann's constant, $T$ is the temperature, and $\kappa$ is the surface curvature, given by $\kappa = 1/r_1 + 1/r_2$, where $r_1$ and $r_2$ are the principal radii of the curvature. Note that the driving force for the surface diffusion in Eq. (1) is the gradient of the surface curvature $\kappa$, which is also the key parameter controlling the Gibbs-Thomson effect. The coefficient $D_s$ for surface diffusion is thermally-activated according to $D_s = D'_s \exp(-Q_s/kT)$, where the pre-factor $D'_s$ and the diffusion barrier $Q_s$ mainly depend on the type of migrating atom

in the CF and on the host dielectric materials. For simplicity, we assume that the CF shape is obtained from the revolution of a curve $\rho(z)$ as shown in Fig. 2f. The normal distance $dn$ traveled by a surface element $\partial s$ during the time increment $dt$ is thus given by[33],

$$\frac{dn}{dt} = -\frac{B}{\rho}\frac{\partial}{\partial s}\left(\rho\frac{\partial \kappa}{\partial s}\right) \qquad (2)$$

where $B = D_s \gamma \delta^4/kT$ is a parameter which strongly depends on the type of CF atom, on temperature and on the surrounding host materials.

Figure 3 shows the simulation results for the morphological evolution of two Ag CFs with the same length $h = 10$ nm and different initial diameters, namely $d_0 = 2$ nm (Fig. 3a) and $d_0 = 0.4$ nm (Fig. 3b). More details about the simulation method and results are reported in the Supplementary Fig. 4–7, Supplementary Note 3 and Supplementary Movies 3–8. The parameter $B$ for Ag is estimated to be $10^{-34}$ m$^4$/s at room temperature (Supplementary Note 4–5 and Supplementary Table 1). Surface diffusion leads to a segmentation of the CF and the formation of one or more intermediate clusters, as shown in Fig. 3b for $t = 1$ ms (see also Supplementary Movie 4 and Supplementary Movie 7). A similar ovulation effect[33] has been recently reported by in situ TEM observations[17,26,34] as the by-product of Ag nanofilament clustering in an insulating material. Note that the specific

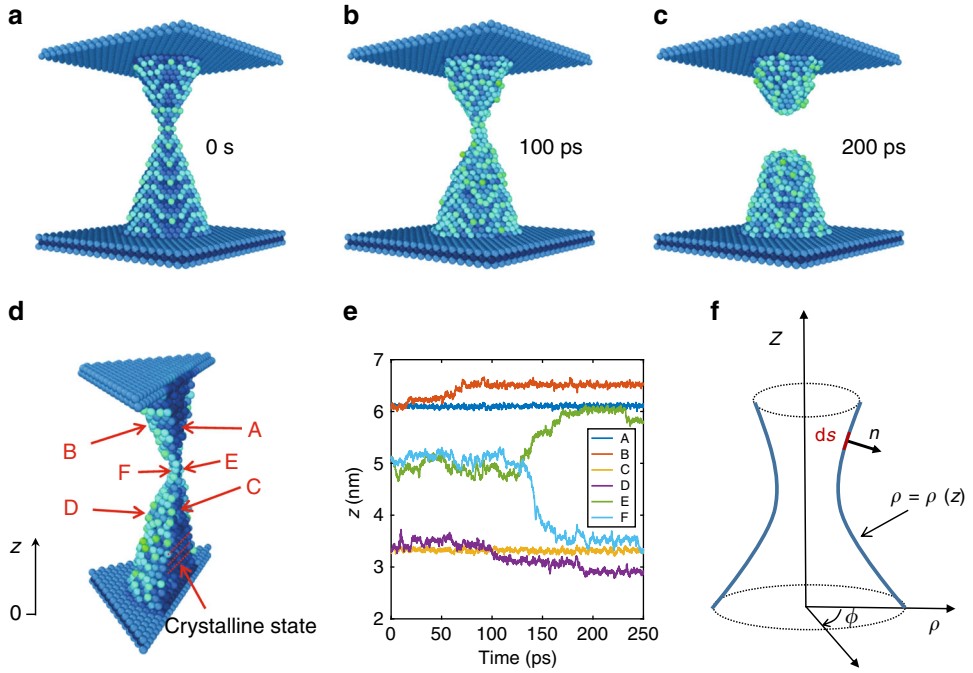

**Fig. 2** MD simulation of surface diffusion. **a–c** MD simulation results of a Ag nanoscale CF between two electrode plates at increasing times, namely $t = 0$, 100 ps, and 200 ps (Supplementary Movies 1–2). The color indicates the free energy per atom with lighter blue representing atoms with higher free energy, hence higher diffusivity. To accelerate the surface diffusion process, the system temperature is set to 800 K, which is still below melting temperature of Ag. **d** Cross-section view of the CF at 100 ps, indicating stable bulk atoms with low free energy and crystalline states, hence negligible diffusion, whereas surface atoms show higher energy and diffusivity, thus resulting in the morphological evolution and rupture of the filament. **e** Typical traces of individual atoms along z direction for bulk atoms (A, C), surface atoms (B, D), and bottleneck atoms (E, F) in **d**. Bulk atoms show constant positions, while the surface atoms migrate toward their closest electrodes. Atoms in the CF bottleneck region also show a high tendency to migration to either the top or bottom electrode. (see Supplementary Note 2 and Supplementary Fig. 2 for more analysis). **f** CF geometry considered in the model of the surface diffusion. For simplicity, isotropic surface diffusion and axisymmetric (along z) CF surface are assumed. $\rho(z)$ defines the geometry of the filament surface by rotating around the z axis, and n defines the surface evolution of a surface piece ds in unit time in outward normal direction

morphological evolution of the filament is affected by the hydrophobicity between filament and electrode materials, initial filament shape (Supplementary Note 6, Supplementary Fig. 8–10 and Supplementary Movies 4–8), and anisotropy of the filament crystal. However, the generic isotropic simulation shown in Fig. 3 can provide us a general rule of the size-dependent stability of the CF.

**Size-dependent lifetime**. We define the lifetime $\tau$ of the CF as the first opening of a depleted gap according to the numerical solution of Eq. (2) (see Fig. 3). Reducing the diameter of the CF by 5 times (from Fig. 3a to Fig. 3b) results in a reduction of the lifetime by about 150 times, as shown by the simulated gap length in Fig. 4a, due to the larger surface curvature $\kappa$ controlling the surface diffusion rate (Eq. (1)) and the CF morphological evolution rate (Eq. (2)). The dependence of lifetime on the initial CF diameter can explain the transition from volatile switching (Fig. 1e) to non-volatile switching (Fig. 1d) at increasing $I_C$, where a larger compliance current leads to a larger CF diameter with longer lifetime. To highlight the size-dependent lifetime, Fig. 4b summarizes the CF lifetime as a function of the initial CF diameter $d_0$ with a constant CF length $h = 10$ nm being assumed. The lifetime increases with the CF size according to a power law $\tau \sim d_0^4$, then increases asymptotically as it approaches a stable size $d_0 \approx 14$ nm. The power law with exponent 4 in our surface-diffusion model arises from Eq. (2), where the filament surface evolution rate $dn/dt$ is proportional to the second-order derivative of the curvature $\kappa$ in Eq. (2), which in turn is a second-order derivative of the surface profile $\rho$ in Eq. (1) (based on similar analysis, we also exclude other possible

mechanism, see Supplementary Note 7). The power law for thin CFs ($d_0 < 5$ nm) is consistent with the Herring's law[30] $s_\tau = \lambda^4$, where $s_\tau$ is the scaling of lifetime with the shape dimension scaling of $\lambda$, thus describing the lifetime $\tau$ as a function of the size of equally-shaped particles (see Supplementary Note 8). Note in fact that the fixed length has negligible effect on the lifetime for $d_0 << h$. The steep increase for higher $d_0$ marks the onset of the CF stability, hence non-volatile switching, which corresponds to the equivalence of the two principle radii of surface curvature (Fig. 4b inset), similar to the liquid capillary bridge between two plates[35].

Results in Fig. 4b provide quantitative criteria for discriminating volatile and non-volatile RS depending on the CF size. For instance, a one-atom-wide CF with $d_0 \approx 0.2$ nm shows a lifetime $\tau \approx 10$ µs, resulting in volatile switching for possible applications as selector device in crossbar arrays or neuromorphic elements. On the other hand, the simulation results project that a CF with $d_0 \approx 14$ nm can have a lifetime $\tau \approx 5 \times 10^8$ s, corresponding to about 16 years at room temperature, which is thus compatible with non-volatile memory applications.

The calculated quantitative timescale of the geometrically identical shape evolution process identified by $d_0/h$ can be extrapolated by Herring's scaling law to interpret some general observations. For instance, a filament with initial diameter $d_0 = 80$ nm and filament length $h = 4$ µm will have the same shape evolution (e.g., resulting in egg-shaped cluster segmentation) of a filament with initial diameter $d_0 = 0.2$ nm and filament length $h = 10$ nm, except for their dimension scaling of $\lambda = 400$ and lifetime scaling of $s_\tau = 2.56 \times 10^{10}$. As a result of the dimensional scaling, the lifetime scales from 10 µs for the filament with

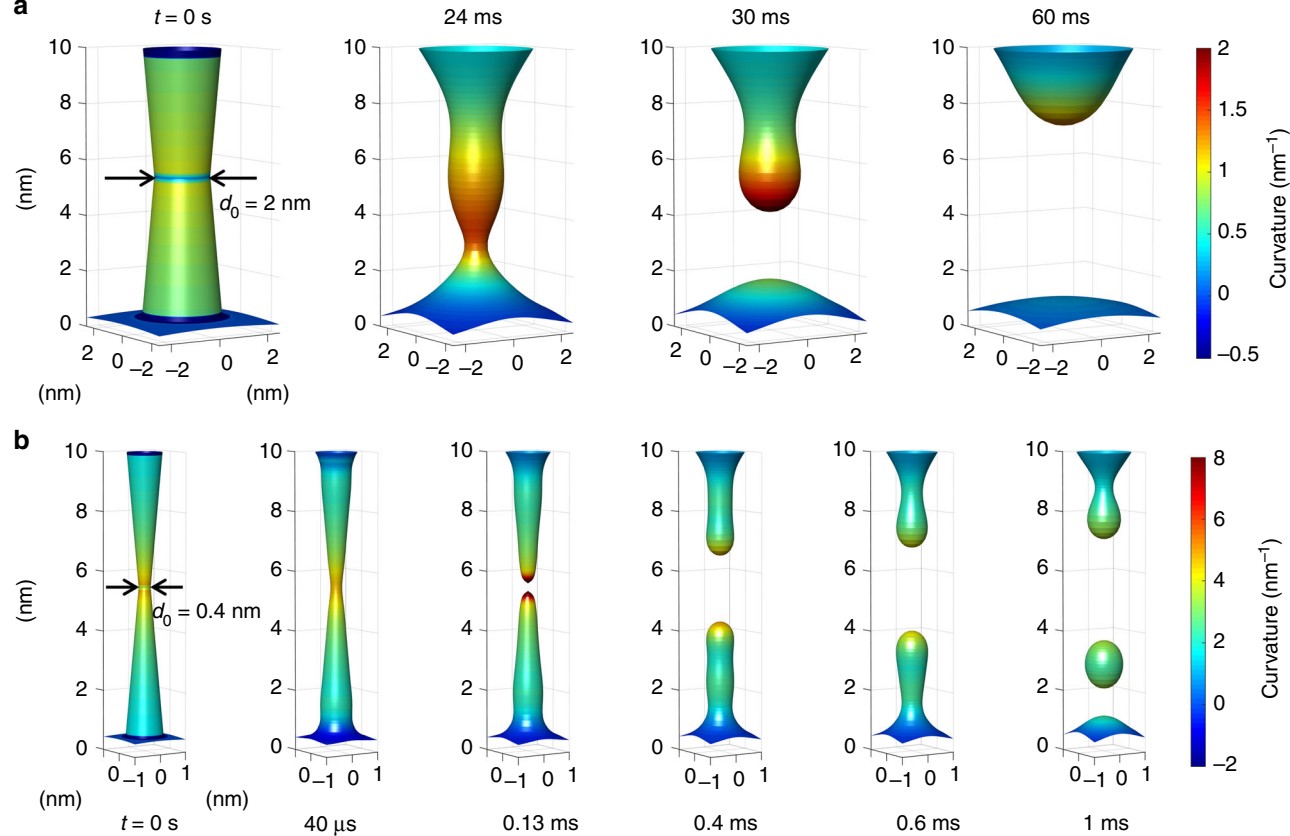

**Fig. 3** Morphological changes induced by surface diffusion. **a** Simulation results of the surface diffusion equations, indicating the rearrangement of a Ag CF with initial diameter $d_0 = 2$ nm at increasing times, namely $t = 0$, 24 ms, 30 ms, and 60 ms. **b** Same as **a**, but for a CF with an initial diameter $d_0 = 0.4$ nm at times $t = 0$, 40 µs, 0.13 ms, 0.4 ms, 0.6 ms, and 1 ms. For both cases, the CF length is 10 nm, equivalent to the thickness of the insulating layer between two planar electrodes. The color in the figures indicates the surface curvature, where surface points with higher curvature are more likely to induce surface diffusion, thus providing a suitable spot for CF rupture. The CF lifetime decreases from around 20 ms (**a**) to about 0.13 ms (**b**) as $d_0$ is reduced from 2 nm to 0.4 nm. The ovulation effect in **b**, where nano-clusters result from the disconnection of a thin CF, is consistent with in situ TEM observations of Ag nano-clusters evolving from Ag nanoscale CFs[17,26,34]

$d_0 = 0.2$ nm and $h = 10$ nm, to $2.56 \times 10^5$ s (about 3 days, in line with the measured lifetime of AgNWs[36]) for the filament with $d_0 = 80$ nm and $h = 4$ µm.

**Universal model of volatile and non-volatile switching**. To further validate our model, we conducted time-resolved experiments to measure the lifetime of Ag CFs as a function of $I_C$. The CF was obtained by applying an excitation pulse at relatively large voltage, followed by a monitor voltage at relatively low voltage $V_{read} = 0.1$ V to monitor the conductance of the device in real time. Figure 5a shows the setup of applied voltage pulse, while the inset shows the measurement configuration, consisting of a RS device connected to a load resistance $R_L$ in serial. The $R_L$ aimed at limiting the current during the set transition. By changing the value of $R_L$ from 1 MΩ to 100 Ω, the CF size could be varied to study the size-dependent lifetime. Figure 5b shows the monitored conductance for various $I_C$ from 1 µA to 10 mA, where the initial conductance $G_0$ shows a linear increase with $I_C$ in agreement with field- and temperature-controlled switching in RS devices[24]. The variable initial conductance is mapped into an initial CF diameter $d_0$ according to $G_0 = \pi\sigma d_0^2/(4h)$, where $\sigma$ is the CF conductivity considering the enhanced scattering effects in nanoscale CFs[37,38] (Supplementary Note 9 and Supplementary Fig. 11). The figure also shows the calculated conductance evolution $G(t)$ from Eq. (2), indicating a good agreement with data. Figure 5c shows the measured and calculated lifetime as a function of $d_0$, also

reporting data of sub-10 nm scale filaments for both Cu[39–43] and Ag[17,23,44,45] from the literatures (more details in the Supplementary Table 2–3). Compared to Ag, Cu-based RS devices show a stronger tendency to non-volatile switching due to its higher surface activation energy $Q_s$[46] and, hence, smaller $B$ (estimated as $10^{-47}$ m⁴/s at room temperature, see Supplementary Note 4). Note the large variation of lifetime, which might originate from anisotropic surface diffusion along different crystalline directions, or inaccurate control of the CF size by $I_C$[47]. Different types of dielectric materials would also impact on the lifetime by affecting parameter $B$ in Eq. (2). Despite these variations, the surface diffusion model accounts for the size dependence of lifetime over a broad range of experimental conditions and samples. The temperature dependent lifetime also relies on the surface activation energy $Q_s$. For the estimations of the parameter $B$ and the predicted size-lifetime lines at different temperatures in Fig. 5c, the surface activation energy $Q_s$ for Cu and Ag was assumed to be 1.1 eV and 0.52 eV, respectively. This is consistent with the data projected from the temperature dependent retention times in Cu and Ag RS devices by fitting with the Arrhenius rule, which gives the value of $Q_s$ in the range of 1.3 ~ 1.4 eV and 0.5 ~ 0.7 eV for Cu and Ag, respectively[43–45].

**Discussion**

Surface self-diffusion effect was first observed in vacuum tube electronic devices, as responsible for the blunting of a sub-mm

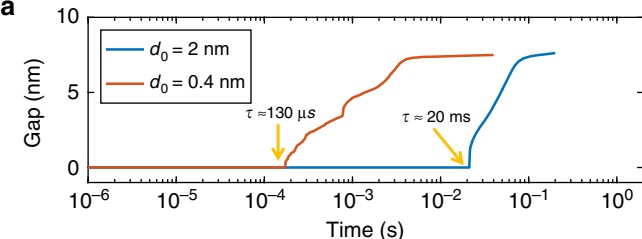

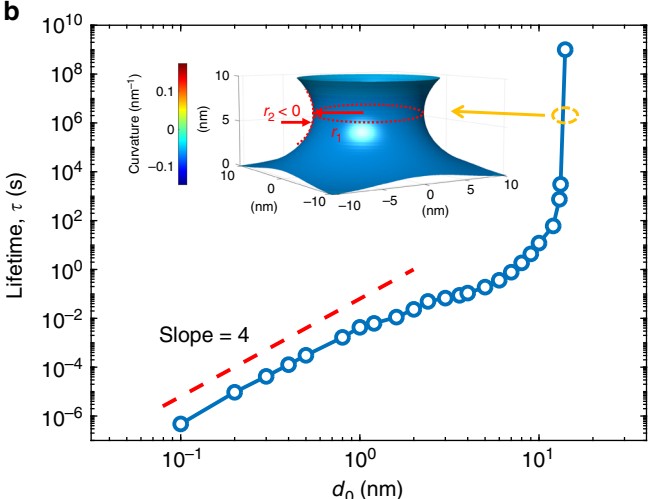

**Fig. 4** CF size-lifetime scaling law. **a** Filament gap length as a function of the relaxation time. The increase of the gap length from zero marks the moment of CF disruption which we defined as lifetime $\tau$. **b** Filament disruption time as a function of initial filament diameter $d_0$. The size-time scaling for thin filament ($d_0 \ll 10$ nm, $h = 10$ nm) well agrees with Herring's law $\tau \sim d_0^4$ with a slope of 4, while, on the other hand, for thick filament ($d_0 > 10$ nm), the lifetime more rapidly increases with the lateral size as a stable capillary bridge is formed, marking an increase of the lifetime from microseconds to years with increasing filament diameter. Inset: snapshot of a stable capillary bridge formed from filament with initial diameter $d_0 = 14$ nm. The stability of this shape originates from the two principle radii of surface curvature having equal modulus and opposite signs. In the bottleneck point, the horizontal radius of the surface curve $r_1 > 0$, whereas the vertical radius $r_2 < 0$, resulting in the surface curvature $\kappa$ tending to zero.

filamentary field-emission cathode operating at elevated temperature (~1000 K)[33]. The lifetime of this type of filament, which controls the vacuum tube lifetime, is generally in the range of hundreds of hours. At nanoscale dimensions, the morphological changes driven by the minimization of surface energy are highly accelerated to the range of observable time scale at room temperature, which is at the origin of a liquid-like pseudoelasticity[31] observed in situ. Surface diffusion effects also control general properties in nanoscale world, such as the lack of sharp tipped metal coated probes for atomic force microscopy[48] compared to the covalent bonded crystallized silicon or diamond probes[49], and the instability of ultrathin metallic NWs, e.g., Ag NWs with diameter less than 40 nm[36].

In this work, we provide evidence for surface diffusion being the fundamental mechanism for the filament rupture in RRAM devices. Other works have previously proposed that out-diffusion, rather than surface diffusion, acts as the leading dissolution process in other materials systems, such as Ni in NiO[50]. We note, however, that out-diffusion is not expected to play a major role in the materials systems considered in our work, namely Ag or Cu in silk and oxide compounds. This is independently supported by at

least two evidences: first, recent TEM observations reveal the presence of Ag or Cu clusters rather than homogeneously distributed Ag within the doped silicon-oxide layer. This was shown for Ag-doped $SiO_x$ ($x < 2$)[17], Cu-doped $SiO_x$ ($x < 2$)[26], and Ag-doped $SiO_2$[27,34]. All these results indicate that Ag and Cu have relatively low solubility in silicon oxide, thus resulting in the clustering of Ag or Cu atoms. The low solubility of Ag and Cu in the host materials can be understood as the result of the low reactivity of Ag and Cu elements and the strong chemical stability of Si-O valence bond (see Supplementary Note 7 for extended discussion). Second, experimental data are consistent with Herring's scaling law $\tau \sim d_0^4$, while one would expect a dependence $\tau \sim d_0^2$ for out-diffusion as reported by our analysis in the Supplementary Note 7, in agreement with previous results[50,51]. Out-diffusion might be non-negligible when the metal has a relatively large solubility in the host materials, such as Ag in $Ag_2S$[52] and Cu in $CuS$[53].

The volatile switching in RS devices has previously been interpreted as the consequence of the atomic clustering to minimize the Gibbs-Thomson energy at the interface between the filament and the host material[17–19,54]. Our interpretation of filament shape evolution induced by surface diffusion also has roots in the Gibbs-Thomson effect, given the dependence on surface curvature radii in Eq. (1). Another common phenomenon induced by Gibbs-Thomson effect is Ostwald ripening[55–58], which has been proposed to control the evolution of the particles obtained by filament fragmentation[17]. According to our model, surface diffusion might control the initial stages of the filament disconnection, which also dictates the filament lifetime according to the Herring's law. Ostwald ripening instead may be responsible for the post-lifetime evolution of the filament particles. In any case, the driving force for the surface diffusion and Ostwald ripening both can be traced back to the Gibbs-Thomson effect (see Supplementary Note 10 and Supplementary Fig. 12 for more discussion).

In RS devices, CFs of various sizes can be electrically formed, resulting in a large range of electrically measurable lifetime. According to our surface diffusion model, the ultimate stability of the CF for non-volatile switching arises from a stable capillary bridge between the top and bottom electrodes. Note that this condition might result in a relatively large CF, which conflicts with other requirements of RS devices for practical applications in non-volatile memories. For instance, RS devices should also be easily erasable at relatively low current, which is necessary to minimize the IR voltage drop across lines in crosspoint arrays. An excessive voltage drop, in fact, would lead to unwanted disturbs to unselected cells. The tradeoff between small operating currents and CF stability requires careful device and materials optimization, toward the minimization of the free energy at the CF surface. Note that experimental data show no tradeoff between endurance and operation current (see Supplementary Note 11 for more discussion). Conversely, minimization of the lifetime in the range of few ns might enable the operation of RS device as fast and efficient selector device for crosspoint arrays[20,59]. The fast recovery of the off-state is essential in this case to enable random access within the array, where each memory must appear unselected immediately after access for read or set/reset. Materials engineering should guide the selection of CF materials for selector technology to enhance the surface energy and the related surface diffusion effects[19].

In summary, we propose a universal surface diffusion mechanism for the spontaneous rupture of metallic CFs in filamentary RS devices. Surface diffusion consistently accounts for the transition from volatile to non-volatile switching observed in nanoscale RS devices, where the lifetime of sub-10 nm CFs can span from microseconds to years. Results provide a general

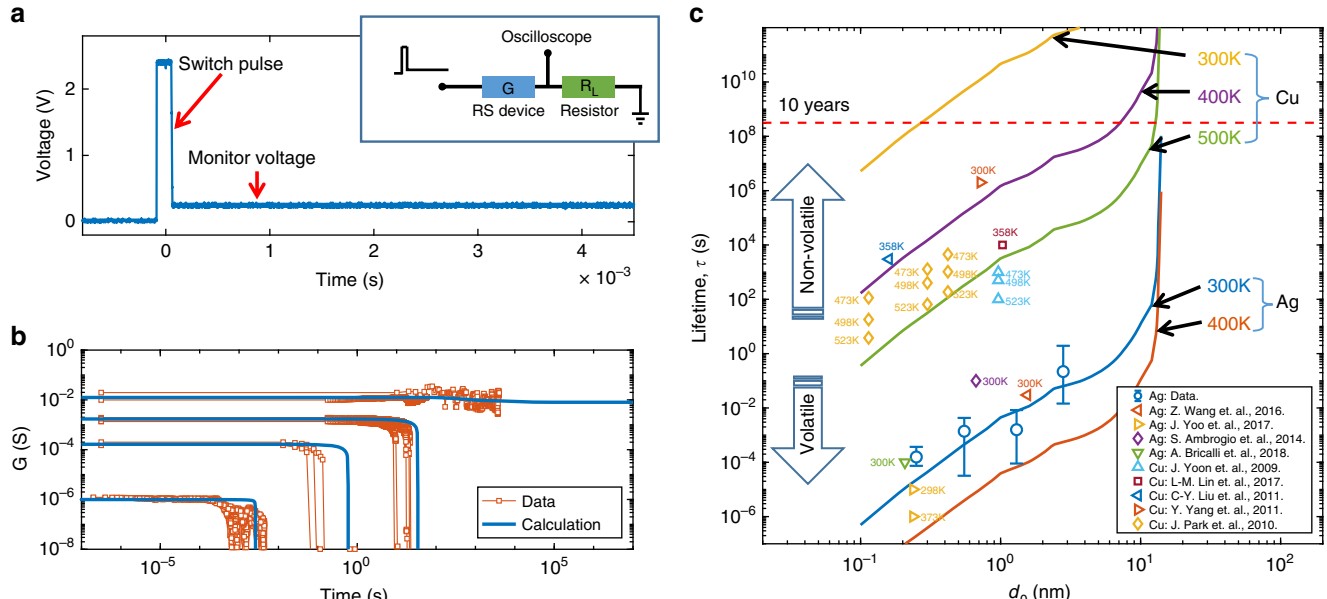

**Fig. 5** Universal explanation for volatile and non-volatile switching. **a** Illustration of the experimental setup for the measurement of the lifetime of Ag filament. The Ag NWs resistive device with conductance $G$ is serially connected with a load resistance $R_L$ which limits the current to a maximum value $I_C$. A voltage pulse (2.5 V, 100 μs) is applied to create the Ag filament, followed by a relatively low monitor voltage (0.1 V) to probe the filament evolution. **b** Measured and calculated conductance for the resistive switching device for different filament sizes. As $I_C$ increases, the lifetime of the Ag filament increases as a result of the size-dependent surface diffusion. The largest $I_C$~10mA results in a stable conductance, corresponding to a stable capillary bridge with a large diameter. **c** Measured and calculated CF lifetime for devices with electrode made of Cu[39–43] and Ag[17,23,44,45] as a function of the initial diameter $d_0$. At room temperature, the parameter $B$ was estimated as $10^{-34}$ m⁴/s and $10^{-47}$ m⁴/s for Cu and Ag filaments, respectively. The values of the surface activation energy $Q_S$ was estimated as 1.1 eV and 0.52 eV for Cu and Ag, respectively. Calculations at various temperatures are also shown. The results account for the transition from volatile switching to non-volatile switching in RS devices (Fig. 1d–f) based on the size-dependent lifetime

framework for understanding the stability of nanoscale structures, and designing RS devices for a wide range of applications, e.g., non-volatile memories with high stability for digital data storage, volatile switching for selector devices in crosspoint arrays, and tunable-lifetime RS synapses for neuromorphic computing with short/long-term plasticity.

## Methods

**Device fabrication and characteristics**. Silver nanowires (length: 10 μm, diameter: 60 nm; concentration: 0.5% wt) were purchased from Sigma-Aldrich. The aqueous solution of silk fibroin for fabricating the device was prepared according to the reported method[20]. Firstly, a 5/70 nm-thickness Cr/Au layer was deposited on silicon substrate as the bottom electrode. Then, 0.1 ml Ag NWs solution was added into 1 ml silk fibroin solution (2% wt) to form the blended solution. The resultant solution was spin-coated onto the bottom electrode at 1000 rpm for 45 s, and then evaporated for 2 h at room temperature. Finally, a 70 nm-thickness Au pad with the size of 100 μm × 100 μm was evaporated as the top electrode. For this device, the effective active electrodes are equivalent to two adjacent AgNWs in the composite film or one Au electrode and one Ag NW electrode. Keithley 4200 semiconductor parameter analyzer was employed to measure the DC electrical characteristics. Arbitrary waveform generators (Agilent 33220A) and oscilloscope (Tektronix DPO5054B) were used for lifetime measurement under pulse mode.

**MD simulation**. The LAMMPS program[28] was used to perform the MD simulations. The interactions between Ag atoms were described by the EAM potential[29]. We used Nose-Hoover NVT ensemble and the system temperature was set to 800 K to observe the fast filament change in a reasonable simulation time. The MD simulations are carried out in an approximate 8 × 8 × 9 nm³ box, with periodic boundary conditions in $x$ and $y$ axes and fixed boundary condition in $z$ axis. Two/three fixed layers of Ag atoms in the top/bottom of the box act as top/bottom electrode. The initial filament shape is that of two-truncated cones (Fig. 2a and Supplementary Movies 1) or cylinder (Supplementary Movies 2), both of which are cut from an FCC Ag crystal.

**Numerical simulation of the filament shape evolution**. The simulations of morphological evolutions were performed using Matlab©R17b with self-design codes. Isotropic surface diffusion and the CF surface of revolution by Eq. (2) were assumed. The morphological evolutions of the CF geometry are simulated in dimension-less form[33]. To this purpose, we defined dimension-less variables $H = h/l$, $D_0 = d_0/l$, $K = \kappa l$, and $\Gamma = Bt/l^4$, where $h$ is the CF length, equivalent to the thickness of the insulating layer in the RS device, $d_0$ is the initial CF diameter, $t$ is time, and $H$, $K$, $\Gamma$ are their dimensionless representations. The constant $l$ (unit: $m$) is the scaling ratio between the CF size and the dimensionless one, defining the spatial dimension of the filament. For each step, the curvatures of the filament surface for each segment were calculated, thus the changes of the morphological profile were obtained from Eq. (2) and the surface profile was updated accordingly. A finite differential method was used to calculate the differential values of surface curvatures and morphological change rates. The output of the computation was converted back to real space/time variables from dimensionless variables. For instance, the real-time value can be obtained by $t = \Gamma l^4/B$. The source code is available from the authors upon request.

## Data availability

The data that support the findings of this study are available from the corresponding author upon reasonable request.

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

## Acknowledgements

This article has received funding from the European Research Council (ERC) under the European Union's Horizon 2020 research and innovation programme (grant agreement No. 648635), Singapore Ministry of Education (MOE2014-T2-2-140 and MOE2015-T2-2-60).

## Author contributions

W.W. proposed the idea and conducted the simulations. M.W. and X.C. conducted the experiments. E.A., A.B., M.L., Z.S., and D.I. assisted the modeling and simulations. W.W. and D.I. wrote the first draft. All of the authors discussed the results and contributed to the preparation of the manuscript. D.I. supervised the research.

## Additional information

**Competing interests:** The authors declare no competing interests.

