## [Peer Review File · Nature Communications]

Reviewers' comments:

Reviewer #1 (Remarks to the Author):

This work proposes a generalized model accounting for the varied lifetime of metallic nanofilaments in resistive switching devices, by identifying self-diffusion of atoms along filament surface as the dominant factor. The varied filament size, practically controlled by electrical operation conditions especially the current compliance, leads to different lifetimes before the device spontaneously switches back to off state, and such phenomenon was explained and fitted by the proposed surface diffusion limited model. The new understanding could be helpful for on-demand optimization of device retention characteristics to meet the different requirements in memory, neuromorphic and selector applications. This reviewer finds the overall concept of the work interesting, but to me the proposed ion transport kinetics and filament model are somewhat over-simplified, as detailed below. A more comprehensive investigation on the properties of host materials and filaments is therefore needed to justify the conclusion.

1. The relative contribution of surface diffusion and out diffusion might be an issue to clarify. The contribution of out diffusion is significantly affected by the property of host materials, such as ion solubility, and the concentration of the same ions (Ag, Cu) already existing in the host materials, etc. Note that the TEM observations cited are from similar type of dielectrics (Refs. 17 and 26). In certain conditions, it might be possible for the out diffusion to become dominant during the dissolution of filaments. In the present study, the consideration on host materials is limited to their impacts on interface conditions and surface diffusions rates. It may therefore be helpful to further check the dependence of out diffusion on host materials, e.g. solid electrolytes with high solubility of ions, to strengthen the discussions.
2. The modeling of the present work was conducted on filaments with roughly symmetric and uniform geometry. However, it has been experimentally observed that strongly asymmetric filament shapes, e.g. cone-shaped filaments, widely exist, where the dissolution of filaments is expected to take place at a critical region. It is also a well-established approach to use bilayer switching dielectrics or asymmetric electrodes in resistive switching cells, where the conducting filament usually consists a robust base and a weak tip, once again confining the dissolution position. In such context, the surface diffusion around the critical position can be enhanced along specific directions given the strong asymmetry, and the relative contributions of surface diffusion and out diffusion may also be changed. Such case is more realistic and represents state-of-the-art resistive switching devices, and hence it will be interesting to check the proposed picture in such type of filament geometries and models.
3. This study plots the lifetime dependence on filament size by considering the filament geometry with a variable of d/h (Fig. 4b). In this case, a filament length of 10 nm was assumed, but typical filament length could have a large variation, ranging from ~ 1 nm (if the filament has a critical switching region) to tens of nm. Will the variation in filament length significantly impact the results, or they will collapse into a similar curve? The authors are recommended to check on this.
4. The device in the present work has an overall symmetric structure, since the Ag atoms are introduced into the dielectric layer instead of coming from one of the electrodes. As a result, it is surprising to see the bipolar switching characteristics in Fig. 1d. A reason or statistic might be needed for that.
5. A minor problem exists in Fig. 5c, where lifetime exceeding 1 second is marked as nonvolatile. It may be inappropriate.

Reviewer #2 (Remarks to the Author):

The manuscript "Surface diffusion-limited lifetime of silver/copper nanofilaments in resistive switching devices" demonstrates simulations as well as experimental validation for filament lifetimes in Resistive switching devices (ReRAM or CBRAM). The manuscript is clear, to the point and explains for the first time on an atomic scale what happens when and how conductive paths are broken.

Using their simulation and model, the authors were able to predict lifetimes of CF for a wide range of diameters and sizes. What is more, the authors experimentally validate those lifetimes, not only for their own silver filament-based devices but also using previously demonstrated work from other groups.

This work is highly relevant for memory-, access device- and neuromorphic computing- research fields and significantly aids the design of devices such that specific characteristics are obtained. The trade of between life-time and energy for forming and erasing a memory state is thus clearly shown. This relation is crucial for optimizing devices for specific applications.

I do have a couple of (minor) questions/comments:

1. Can the authors be sure that the distance between AgNW is similar each time in the experimental data? The compliance current is used to control the final thickness of the CF. In the simulations, this distance (between top and bottom electrodes) is set to a certain value (10 nm). However, in the experimental data, a number of AgNW (randomly) exist in the silk dielectric. It seems that once a filament has formed, this will draw all the current (and perhaps still grow in diameter) and no other filaments will form, is that correct? And if so, the CF will not always grow between similar (electrode/AgNW) distances as is modelled, won't that affect the comparison with the model?

2. If I understand correctly, the MD simulation shows how the atoms diffuse which indicates that energy minimization is responsible for that, either driven by the gradient of atomic vacancy concentration or chemical potential. Using this knowledge, a differential equation is presented that describes this relation which in turn is used for the second, morphological model. Is MD in any way also used to verify the other model, for instance having an identical CF with similar dimensions of height and diameter for the two models as presented in Fig 2 (MD) and Fig 3 (Morphological)? That could give a direct verification of the second model.

3. Some of the current issues with memristive (ReRAM) devices, such as non-linear conductance tuning and stochastic behaviour, are hampering efficient operation in neuromorphic applications, so I am wondering whether this model can also be used to optimize forming and breaking conditions as such that linear/analogue tuning of the conductance is achieved/improved? I could imagine the correct voltage and current pulses necessary to achieve a certain conductance can be predicted, resulting in accurate tuning without read-actions between the tuning (which significantly decreases the efficiency of those neuromorphic arrays). For some neuromorphic applications for instance, only a relatively short life-time (~mins) is necessary as long as the conductance tuning is done efficiently and predictable. Can the authors comment on that?

4. Finally, would it also be possible to use this model to calculate the exact energy necessary for switching? That could also be a valuable metric to know, to design devices for (energy) efficient memory and neuromorphic applications.

Reviewer #3 (Remarks to the Author):

The manuscript is a very interesting account on molecular dynamics modelling of Ag conductive filaments and switching data on nanorods embedded in a silk matrix sandwiched between Au electrodes. The key message is that the lifetime of the device "ON" state, that is, the device low resistance state, is governed by the size of the filament. The filament size can be controlled by the compliance current, presumably due to some Joule heating/electromigration cooperative phenomena. The key point for the explanation of device performance is given by surface diffusion, and several movies corresponding to simulations of conductive filaments of various diameters show a power law dependence in which the lifetime scales with the fourth power of the filament diameter. This in turn is in agreement with Herring's law. This all sits very well together, and seems to explain well the rupture phenomena.

I do have a couple of points to make though. Surface diffusion and Herring's law seem to explain the observed phenomena, but I would think that the Gibbs-Thomson equation leading to Ostwald ripening phenomena could bear similar results. In Ostwald ripening, the critical issue is the radius of curvature, the smaller the less stable. To be more specific, the vapor pressure of the nanoparticle is inversely proportional to its radius, and therefore the atoms just want to leave the particle. From that perspective, and in qualitative terms, the manuscript does not add anything new; I could say that the observed results are indeed to be expected.

My first point then is: could we explain the observed phenomena by the Gibbs-Thomson equation, and what is missing from that formalism to properly capture the observed results? Very carefully pointed out is that in matrices, the surface (or should it be interface?) diffusion processes should change. Could we still talk about universal behavior? How does that fit in with Ostwald-ripening (see for example [1])?

My second point is: I am assuming that the devices can switch multiple times. From the learnings of this work, and knowing that the devices "die" most often in the "ON" state, can one find a compromise between endurance and retention?

In summary I find the paper appropriate for Nature Comms, however to have a more general impact, the Gibbs-Thomson phenomenon should be discussed/ruled out/accommodated so the reader can connect different fields and gather a more broad perspective. From a more area specific impact, I would like to see some discussion on retention x endurance. Could this work shed some light into this aspect? Are those two issues inextricably connected? After comments from the authors addressing these two points I find the paper suitable for publication.

[1] see, for example: [dx.doi.org/10.1021/ja309034n](https://doi.org/10.1021/ja309034n), Simo et al., J. Am. Chem. Soc. 2012, 134, 18824–18833

Reply to the comments of Referee #1:

- *This work proposes a generalized model accounting for the varied lifetime of metallic nanofilaments in resistive switching devices, by identifying self-diffusion of atoms along filament surface as the dominant factor. The varied filament size, practically controlled by electrical operation conditions especially the current compliance, leads to different lifetimes before the device spontaneously switches back to off state, and such phenomenon was explained and fitted by the proposed surface diffusion limited model. The new understanding could be helpful for on-demand optimization of device retention characteristics to meet the different requirements in memory, neuromorphic and selector applications. This reviewer finds the overall concept of the work interesting, but to me the proposed ion transport kinetics and filament model are somewhat over-simplified, as detailed below. A more comprehensive investigation on the properties of host materials and filaments is therefore needed to justify the conclusion.*

Reply: We thank the Referee for the insightful comments on our work. All the Referee's comments have been very helpful to improve the presentation of our findings. Following the Referee's concerns and suggestions as well as other Referees' comments, we have extended our study to make the work more comprehensive and fully support our conclusions. More detailed answers to the Referee's comments are reported in the following.

- *1. The relative contribution of surface diffusion and out diffusion might be an issue to clarify. The contribution of out diffusion is significantly affected by the property of host materials, such as ion solubility, and the concentration of the same ions (Ag, Cu) already existing in the host materials, etc. Note that the TEM observations cited are from similar type of dielectrics (Refs. 17 and 26). In certain conditions, it might be possible for the out diffusion to become dominant during the dissolution of filaments. In the present study, the consideration on host materials is limited to their impacts on interface conditions and surface diffusions rates. It may therefore be helpful to further check the dependence of out diffusion on host materials, e.g. solid electrolytes with high solubility of ions, to strengthen the discussions.*

Reply: We agree with the Referee that the contribution of the out-diffusion of filament atoms might be affected by the properties of the host materials. However, we believe that, at least for the material systems considered in our manuscript, namely Ag or Cu in silicon oxide, metal oxide or polymer host materials, surface diffusion, rather than out-diffusion, dominates the filament dissolution. This conclusion is based on the following two points:

- 1) Recent high resolution TEM observations reveal the presence of Ag or Cu clusters rather than homogeneously distributed Ag within the doped dielectric layer^[R1-R4]. This was shown for Ag-doped SiO_x (x<2)^[R1], Cu-doped SiO_x (x<2)^[R2], and Ag-doped SiO₂^[R3,R4]. All these results indicate that Ag and Cu have low solubility in silicon oxide, thus resulting in the clustering of Ag or Cu atoms. For instance, in ref. [R1], TEM observations of Ag-doped SiO_x layer revealed that the as-fabricated device shows Ag clusters in SiO_x rather than homogeneously distributed Ag in SiO_x. The low solubility of Ag and Cu in the host

materials can be understood as the result of (i) the strong chemical stability of Si-O valence bond, and the fact that Ag and Cu are generally inert metal and hard to get oxidized. For instance, according to the metal reactivity series^[R5], metals can be ranked as follows: Al > Ti > Cr > Ni > Pb > Cu > W > Ag > Au, while in the electronegativity (Allen) scale^[R6,R7], the ranking is as follows: Hf (1.16) < Ta (1.34) < Ti (1.38) < W (1.47) < Cu (1.85) < Ag (1.87) < Au (1.92);

- 2) The experimental size-dependent lifetime supports surface diffusion as opposed to out-diffusion as the fundamental mechanism for filament dissolution. The surface diffusion theory predicts a lifetime $\tau \sim d_0^n$ with an exponent $n = 4$, which is consistent with the experimental results in Fig. 5c. On the other hand, one should expect an exponent $n = 2$ for out-diffusion as reported by previous simulation studies^[R8,R9].

We agree with the Referee that out-diffusion might play a non-negligible role in other materials systems, such as when the metallic element (Ag, Cu) has a high solubility in the host materials, e.g., metal-chalcogenide materials, Ag₂S for Ag filament^[R10], or CuS for Cu filament^[R11]. In these cases, a different analysis and numerical approach should be conducted to properly take out-diffusion into account. However, in our experiments, we focused on polymer (silk) as dielectric layer between two AgNW electrodes. Ag atoms are more likely to cluster together rather than diffuse homogeneously as single atoms, given the low solubility of Ag into the host polymer. As a result, silk almost behaves as vacuum in this case, which is what has been simulated in Fig. 2. Similarly, all data collected in Fig. 5 refer to oxide dielectric layers, such as SiO_x^[R1-R4], GdO_x^[R12,R13], HfO₂^[R14], and others, which all share a low solubility with the active metals Ag or Cu. In fact, at room temperature (300 °C), the relative solubility of Ag in SiO₂ is 10⁻⁶, namely three orders of magnitude smaller than in Ag₂S beta phase, where the solubility is 10⁻³^[R15,R16]. Thus, our main finding about the leading role of surface diffusion in controlling the lifetime of RRAM should be restricted to these material systems with relatively-low solubility, rather than solid electrolyte materials with relatively-high solubility.

To clarify this point, we added one paragraph in the section ‘Discussion’ (second paragraph of the section, page 7). In the new paragraph, we describe the additional evidences for surface-diffusion as the fundamental mechanism for retention, as opposed to out-diffusion, namely i) TEM observations of clustering in the as-deposited state, and ii) the different size-dependent law for out diffusion ($\tau \sim d_0^2$ instead of $\tau \sim d_0^4$). We also added a more comprehensive study of size-dependent lifetime for out-diffusion in the Supplementary Information (newly added Part X). References R8-R9 were added accordingly.

[R1] Wang, Z. *et al.* Memristors with diffusive dynamics as synaptic emulators for neuromorphic computing. *Nat. Mater.* **16**, 101–108 (2017).

[R2] Yuan, F. *et al.* Real-time observation of the electrode-size-dependent evolution dynamics of the conducting filaments in a SiO₂ layer. *ACS Nano* **11**, 4097–4104 (2017).

[R3] Yang, Y. *et al.* Electrochemical dynamics of nanoscale metallic inclusions in dielectrics. *Nat. Commun.* **5**, 4232 (2014).

[R4] Zhao, X. *et al.* Breaking the current-retention dilemma in cation-based resistive switching devices utilizing graphene with controlled defects. *Adv. Mater.* **30**, 1705193 (2018).

[R5] Greenwood, N. N. and Earnshaw, A. (1984). *Chemistry of the Elements*. Oxford: Pergamon Press. pp. 82–87.

- [R6] “Electronegativity”, Wikipedia page, url: <https://en.wikipedia.org/wiki/Electronegativity>, access date: Oct. 26, 2018.
- [R7] Mann, J. B., Meek, T. L., & Allen, L. C., Configuration Energies of the Main Group Elements, *J. Am. Chem. Soc.* **122**, 2780-2783 (2000).
- [R8] Aga, F. G. et al. Retention modeling for ultra-thin density of Cu-based conductive bridge random access memory (CBRAM). *AIP Adv.* **6**, 025203 (2016).
- [R9] Ielmini, D., Nardi, F., Cagli, C. & Lacaita, A. L. Size-dependent retention time in NiO-based resistive-switching memories. *IEEE Electron Device Lett.* **31**, 353–355 (2010).
- [R10] Ohno, T. *et al.* Short-term plasticity and long-term potentiation mimicked in single inorganic synapses. *Nat. Mater.* **10**, 591–595 (2011).
- [R11] Sakamoto, T. *et al.* Nanometer-scale switches using copper sulfide. *Appl. Phys. Lett.* **82**, 3032–3034 (2003).
- [R12] Yoon, J. et al. Analysis of copper ion filaments and retention of dual-layered devices for resistance random access memory applications. *Microelectron. Eng.* **86**, 1929–1932 (2009).
- [R13] Park, J. et al. Investigation of state stability of low-resistance state in resistive memory. *IEEE Electron Device Lett.* **31**, 485–487 (2010).
- [R14] Yoo, J., Park, J., Song, J., Lim, S. & Hwang, H. Field-induced nucleation in threshold switching characteristics of electrochemical metallization devices. *Appl. Phys. Lett.* **111**, 063109 (2017).
- [R15] McBrayer, J. D., Swanson, R. M. & Sigmon, T. W. Diffusion of Metals in Silicon Dioxide. *J. Electrochem. Soc.* **133**, 1242 (1986).
- [R16] Sharma, R. C. & Chang, Y. A. The Ag–S (Silver-Sulfur) system. *Bull. Alloy Phase Diagrams* **7**, 263–269 (1986).

- *2. The modeling of the present work was conducted on filaments with roughly symmetric and uniform geometry. However, it has been experimentally observed that strongly asymmetric filament shapes, e.g. cone-shaped filaments, widely exist, where the dissolution of filaments is expected to take place at a critical region. It is also a well-established approach to use bilayer switching dielectrics or asymmetric electrodes in resistive switching cells, where the conducting filament usually consists a robust base and a weak tip, once again confining the dissolution position. In such context, the surface diffusion around the critical position can be enhanced along specific directions given the strong asymmetry, and the relative contributions of surface diffusion and out diffusion may also be changed. Such case is more realistic and represents state-of-the-art resistive switching devices, and hence it will be interesting to check the proposed picture in such type of filament geometries and models.*

Reply: We agree with the Referee that asymmetric filament shapes and asymmetric electrodes are more common and more realistic in state-of-the-art resistive switching devices. We have added more simulation results regarding asymmetric electrodes and filament shape in the Supplementary Information, as detailed in the following.

We extended the simulation analysis in the Supplementary Part V, addressing the effect of different electrode materials. This has been modeled by varying the boundary conditions, namely the contact angle between the filament and the electrode, which is dictated by the wettability of the filament/electrode materials (see Fig. S4b). Simulation results show that changing the electrode materials does not significantly affect the size dependence of the lifetime (Fig. S8a and S8b). On the other hand, when the top electrode is hydrophobic with respect to the filament, no stable capillary bridge can be formed anymore, thus the lifetime follows Herring's law across the whole range of the initial diameter d_0 (Fig. S8c).

In the newly added Supplementary Part VI, we addressed the effect of the different initial filament shapes. To this purpose, we varied the conical angle α , defined in the Supplementary Part III. Similar size dependences of the CF lifetime for various α can be observed in Fig. S9. A cylindrical filament ($\alpha = 0^\circ$) has a shorter lifetime compared to the conical filament with higher angle, e.g. $\alpha = 5^\circ$ and $\alpha = 10^\circ$. However, the general trend of lifetime as a function of the initial filament diameter does not change. From the simulation results in Fig. S9, it can be concluded that the filament dimension (diameter) at the critical position dominates the lifetime of the filament. As a general rule, we may consider the region close to the bottleneck as the active area of the filament, while the largest sections of the filament may serve as electrode extensions. In fact, the electrical measurable parameter, i.e., the conductance of the filament, is generally controlled by the minimum filament diameter in correspondence of the bottleneck. The filament with conical shape tends to break at its bottleneck point and retract to the electrodes (Supplementary Movie 4), while a thin cylindrical shape tends to break at multiple points and evolve to separate spheres (Supplementary Movie 7).

- *3. This study plots the lifetime dependence on filament size by considering the filament geometry with a variable of d/h (Fig. 4b). In this case, a filament length of 10 nm was assumed, but typical filament length could have a large variation, ranging from ~1 nm (if the filament has a critical switching region) to tens of nm. Will the variation in filament length significantly impact the results, or they will collapse into a similar curve? The authors are recommended to check on this.*

Reply: As the filament disconnection generally takes place at the bottleneck, the minimum filament diameter d_0 is the most relevant factor for the filament lifetime. Any variation of the filament length has less impact than variations of d_0 . We addressed the impact of the filament length on lifetime in the newly added Supplementary Part VII and Fig. S10. From these results, it can be seen that a relatively small variation of filament length plays a minor role in the lifetime.

With respect to the original version of the manuscript, we decided to change the axis scale in Fig. 4b, where data are now plotted as a function of d_0 , instead of d_0/h . This is because the original figure might suggest that τ is only a function of d_0/h , i.e., equal d_0/h but different dimension scale would lead to the same retention time. This is of course not correct, since, according to our theoretical analysis in Eqs. (1) and (2) and based on Herring's scaling law, the retention time follows the scaling rule $s_\tau \sim \lambda^4$, where s_τ and λ are the scaling factors of the time and the filament size, respectively. The variable d_0/h only defines the initial aspect ratio of the filament. Assuming a constant d_0/h and scaling the size of the filament by a factor λ , the shape evolution of the filament would be identical, although on a different time scale $s_\tau \sim \lambda^4$. For instance, a filament with initial diameter $d_0 = 0.2 \text{ nm}$ and filament length $h = 10 \text{ nm}$ shows the same evolution process (e.g., fragmentation

in clusters, etc.) as a filament with initial diameter $d_0 = 80 \text{ nm}$ and filament length $h = 4 \text{ }\mu\text{m}$, except for their dimension scaling of $\lambda = 400$ and lifetime scaling of $s_\tau = 2.56 \times 10^{10}$. According to our simulation, the lifetime of the former case is about $10 \text{ }\mu\text{s}$, thus, a silver nanowire with diameter 80 nm will have a lifetime of $2.56 \times 10^5 \text{ s}$ (several days), which is consistent with recent results^[R17]. We added one paragraph in the main manuscript (last paragraph of the sub-section ‘Size-dependent lifetime’, page 6) to explain the use of the variable d_0/h for initial shape identity and the extrapolation to the lifetime for silver nanowires.

[R17] Deignan, G. & Goldthorpe, I. A. The dependence of silver nanowire stability on network composition and processing parameters. *RSC Adv.* **7**, 35590–35597 (2017).

- *4. The device in the present work has an overall symmetric structure, since the Ag atoms are introduced into the dielectric layer instead of coming from one of the electrodes. As a result, it is surprising to see the bipolar switching characteristics in Fig. 1d. A reason or statistic might be needed for that.*

Reply: Our device has a symmetric structure, with both electrodes being Ag nanowires as shown in Fig. 1c. Ag is then introduced by migration from one electrode after the first electrical forming, when a positive voltage is applied to the top electrode thus inducing migration of Ag ions from the top electrode. After forming, the filament might be stable in the dielectric layer, thus can be retracted by ion migration to the top electrode induced by the application of a negative voltage. The filament can be restored by the application of positive voltage to the top electrode, which thus explains the bipolar switching characteristics in Fig. 1d. Similarly, the device can be obtained by forming under a negative voltage, then followed by reset under positive voltage and set under negative voltage, which would lead to bipolar switching with opposite polarity compared to Fig. 1d. To better highlight the possibility for bipolar switching occurring under positive forming and negative forming, we have reported more non-volatile I-V curves in the newly added Supplementary Part I (Supplementary Information, page 2).

It should be noted that such bipolar switching behavior only applies for a stable (non-volatile) filament, which is the case for relatively large compliance current (hence diameter of the filament). On the other hand, the device operated in the volatile switching regime leads to bi-directional switching where the filament formation in the on state can take place by ion migration either from the positively-biased top electrode, or from the positively-biased bottom electrode, as indicated in Fig. 1e.

- *5. A minor problem exists in Fig. 5c, where lifetime exceeding 1 second is marked as nonvolatile. It may be inappropriate.*

Reply: We agree with the Referee that the boundary between nonvolatile and volatile behaviors is not straightforward, as it derives from an assumed criterion for the filament lifetime. To avoid misunderstanding, we decided to delete the dashed line in Fig. 5c and avoid an arbitrary separation between volatile and nonvolatile regimes.

Referee #2:

- *The manuscript “Surface diffusion-limited lifetime of silver/copper nanofilaments in resistive switching devices” demonstrates simulations as well as experimental validation for filament lifetimes in Resistive switching devices (ReRAM or CBRAM). The manuscript is clear, to the point and explains for the first time on an atomic scale what happens when and how conductive paths are broken.*

Using their simulation and model, the authors were able to predict lifetimes of CF for a wide range of diameters and sizes. What is more, the authors experimentally validate those lifetimes, not only for their own silver filament-based devices but also using previously demonstrated work from other groups.

This work is highly relevant for memory-, access device- and neuromorphic computing- research fields and significantly aids the design of devices such that specific characteristics are obtained. The trade of between life-time and energy for forming and erasing a memory state is thus clearly shown. This relation is crucial for optimizing devices for specific applications.

I do have a couple of (minor) questions/comments:

Reply: We thank the Referee for the insightful comments on our work. All the Referee’s comments have been very helpful to improve the presentation of our findings. Following the Referee’s concerns and suggestions as well as other Referees’ comments, we have extended our study to make the work more comprehensive and fully support our conclusions. More detailed answers to the Referee’s comments are reported in the following.

- *1. Can the authors be sure that the distance between AgNW is similar each time in the experimental data? The compliance current is used to control the final thickness of the CF. In the simulations, this distance (between top and bottom electrodes) is set to a certain value (10 nm). However, in the experimental data, a number of AgNW (randomly) exist in the silk dielectric. It seems that once a filament has formed, this will draw all the current (and perhaps still grow in diameter) and no other filaments will form, is that correct? And if so, the CF will not always grow between similar (electrode/AgNW) distances as is modelled, won’t that affect the comparison with the model?*

Reply: We agree with the Referee about the physical picture of filament formation. In our device, Ag NWs are dispersed within the silk dielectric and the filament is randomly formed across an active region where two Ag NWs become very close one to each other, one being in contact with the positive electrode, while the other is in contact with the negative electrode. After forming, all the current flows through the only existing filament, thus preventing the formation of other filaments because of the compliance system which causes limitation of the voltage to maintain a certain compliance current I_C . After the formation, the filament can then be electrically or spontaneously disconnected, which represent the cases of nonvolatile and volatile switching, respectively. In the subsequent set operation, the filament would likely form in the same active region, thus the distance between the Ag NWs remains the same. It is true that each device has a different distance between the active NWs, as a result of the random location of NWs in the host material. In our previous experimental work^[R13], we found that the threshold voltage for switching shows a remarkably uniform distribution, thus suggesting a relatively small

variation of the Ag NW distance at the weakest point. As a result, the assumption of a constant h throughout the simulation seems a reasonable assumption. We added one paragraph in the revised manuscript (last paragraph of the sub-section ‘Volatile and non-volatile switching,’ pages 3 and 4) to clarify this point. It should also be mentioned that the filament lifetime depends only weakly on the filament length h , as reported by simulation results in the newly added Part VII of the Supplementary Information. Therefore, random variations of filament length from device to device should negligibly affect the lifetime.

On the other hand, the filament diameter d_0 at the critical position dominates both the lifetime of the filament and the electrical conductance of the volatile device. More simulation results are provided in the newly added Part VI of the Supplementary Information to support this point. In particular, from the reported simulation results, the filament lifetime is less sensitive to the filament height h than to the bottleneck diameter d_0 , thus supporting the assumption of a constant filament length h in the simulation of our devices.

[R13] Wang, M. et al. Enhancing the Matrix Addressing of Flexible Sensory Arrays by a Highly Nonlinear Threshold Switch. *Adv. Mater.* **30**, 1802516 (2018).

- *2. If I understand correctly, the MD simulation shows how the atoms diffuse which indicates that energy minimization is responsible for that, either driven by the gradient of atomic vacancy concentration or chemical potential. Using this knowledge, a differential equation is presented that describes this relation which in turn is used for the second, morphological model. Is MD in any way also used to verify the other model, for instance having an identical CF with similar dimensions of height and diameter for the two models as presented in Fig 2 (MD) and Fig 3 (Morphological)? That could give a direct verification of the second model.*

Reply: We agree with the Referee’s point that a direct comparison between the MD model and the numerical model might help supporting the physical basis for the latter. Following this suggestion, we simulated the relaxation process of a same filament according to the two models, which is reported in the newly added Supplementary Part IV and Supplementary Movie 3. The filament geometries simulated by the two models show strikingly similar evolutions. The time evolution of the filament gap length according to the two models is also shown in the movie. It should be noted that the MD simulation is extremely time consuming and can only be conducted in practical times by assuming a high accelerating temperature in the simulation (e.g., 800 K in Fig. S7). The parameters for MD simulations are solely calculated from first principle, while the parameters for the numerical model are obtained from the literature and, in part, from the best fitting of the experimental data. For these reasons, the timescales of the two models are not consistent with each other, therefore we assumed a normalized time axis in the comparison of the two simulations in Fig. S7c.

- *3. Some of the current issues with memristive (ReRAM) devices, such as non-linear conductance tuning and stochastic behaviour, are hampering efficient operation in neuromorphic applications, so I am wondering whether this model can also be used to optimize forming and breaking conditions as such that linear/analogue tuning of the conductance is achieved/improved? I could imagine the correct voltage and*

current pulses necessary to achieve a certain conductance can be predicted, resulting in accurate tuning without read-actions between the tuning (which significantly decreases the efficiency of those neuromorphic arrays). For some neuromorphic applications for instance, only a relatively short life-time (~mins) is necessary as long as the conductance tuning is done efficiently and predictable. Can the authors comment on that?

Reply: We agree with the Referee that linear and analogue tuning of the ReRAM conductance is crucial for the understanding and control of the resistive device for both memory and neuromorphic applications.

Unfortunately, the processes of electrical force induced increasing/decreasing the filament conductance are generally governed by the field-induced migration, which is not included in our model at the present stage. We currently have some ideas about how to deal with this limitation, however extending the numerical model to describe field-induced drift in addition to surface diffusion would take quite some time and probably be a challenging project on its own.

In this work, we focused on filament spontaneous disconnection induced by surface diffusion, where the gradient of the surface atomic chemical potential is acting as the primary driving force. From this conceptual view, we may extrapolate that, under an applied voltage and the consequent electric field, the bulk atoms might be still hard to move while the surface atoms can migrate by surface *drift* driven by the gradient of the electrostatic potential, i.e., the electric field. By describing the cooperation and competition between drift and diffusion of the surface atoms, the evolution of filament for memory set/reset processes, or synaptic potentiation/depression, can be predicted by the model, thus allowing to optimize the linear/analogue tuning of the conductance. We are currently extending the model along this line, however the model development and validation with experimental data are still in the early stages.

- *4. Finally, would it also be possible to use this model to calculate the exact energy necessary for switching? That could also be a valuable metric to know, to design devices for (energy) efficient memory and neuromorphic applications.*

Reply: We agree with the Referee that the switching energy is a valuable metric. As mentioned in the previous point, the filament evolution induced by field-induced drift is not yet included in our model. Therefore, a direct calculation of the switching energy $E = V \cdot I \cdot t$ is not yet feasible by the model. However, from the MD simulation, we can extract the system total energy of the initially formed filament (Fig. 2a) and the reduced filament energy after some relaxation time. This energy difference also provides the minimum energy to be provided to the system during the switching operation. The corresponding energy relaxation curve as a function of time was added and commented in Supplementary Part II, where the energy decrease after relaxation is about 65 eV, or about 10 aJ. Note, however, that this is just the free energy difference, or ΔG , rather than the real energy expense needed to accelerate the switching transition to the on state, which will instead depend on the voltage, current, and time to achieve the transition.

Referee #3:

- *The manuscript is a very interesting account on molecular dynamics modelling of Ag conductive filaments and switching data on nanorods embedded in a silk matrix sandwiched between Au electrodes. The key message is that the lifetime of the device “ON” state, that is, the device low resistance state, is governed by the size of the filament. The filament size can be controlled by the compliance current, presumably due to some Joule heating/electromigration cooperative phenomena.*

The key point for the explanation of device performance is given by surface diffusion, and several movies corresponding to simulations of conductive filaments of various diameters show a power law dependence in which the lifetime scales with the fourth power of the filament diameter. This in turn is in agreement with Herring’s law. This all sits very well together, and seems to explain well the rupture phenomena.

Reply: We thank the Referee for the insightful comments on our work. All the Referee’s comments have been very helpful to improve the presentation of our findings. Following the Referee’s concerns and suggestions as well as other Referees’ comments, we have extended our study to make the work more comprehensive and fully support our conclusions. More detailed answers to the Referee’s comments are reported in the following.

- *I do have a couple of points to make though. Surface diffusion and Herring’s law seem to explain the observed phenomena, but I would think that the Gibbs-Thomson equation leading to Ostwald ripening phenomena could bear similar results. In Ostwald ripening, the critical issue is the radius of curvature, the smaller the less stable. To be more specific, the vapor pressure of the nanoparticle is inversely proportional to its radius, and therefore the atoms just want to leave the particle. From that perspective, and in qualitative terms, the manuscript does not add anything new; I could say that the observed results are indeed to be expected.*

Reply: Indeed our surface diffusion mechanism has its basic physical origin in the Gibbs-Thomson equation. In the Gibbs-Thomson effect, the vapor pressure of a nanoparticle increases with the inverse of its radius^{[R18],[R19]}. Ostwald ripening is one example of the Gibbs-Thomson effect, where atoms released by small particles may attach to larger particles, which then grow at the expense of smaller particles. Similar to Ostwald ripening, also our picture for the filament disconnection can be viewed as a consequence of the Gibbs-Thomson effect, which drives the minimization of the surface energy. However, there is a major difference between our model and the Ostwald ripening. In Ostwald ripening, the transfer of atoms from small to large particles occurs via out-diffusion in the gas (or liquid, or solid) phase surrounding the particles. Our model for the filament evolution, instead, is entirely based on surface diffusion, as opposed to out-diffusion. It would cost too much energy for atoms to leave the filament and get dispersed in the host material, especially for relatively low solubility of filament metallic atoms in the host material. Therefore, we believe that, although our model for filament evolution and the Ostwald ripening share the same physical origin, i.e., the Gibbs-Thomson effect, they are totally different processes.

We conclude that, although our model has its physical origin in the Gibbs-Thomson effect, and although some of the simulation results are not unexpected based on the Gibbs-Thomson effect, our model is indeed new, because it allows, for the first time, to predict the time evolution of a filament based on its geometry, materials, and conditions (e.g., temperature). Most importantly, through our model we can derive a universal law bridging together many experimental results in the literature under the same physical picture for the first time.

- *My first point then is: could we explain the observed phenomena by the Gibbs-Thomson equation, and what is missing from that formalism to properly capture the observed results? Very carefully pointed out is that in matrices, the surface (or should it be interface?) diffusion processes should change. Could we still talk about universal behavior? How does that fit in with Ostwald-ripening (see for example [1])?*

Reply: Gibbs-Thomson effect has several equivalent forms. The most popular version of the Gibbs-Thomson effect gives the vapor pressure p at the liquid-vapor interface as a function of the local curvature radius according to^[R18]:

$$p = p_0 - \frac{\gamma \rho_v}{\rho_l - \rho_v} \left(\frac{1}{r_1} + \frac{1}{r_2} \right), \quad (\text{R1})$$

where p_0 is the vapor pressure for a flat surface ($r_1, r_2 = \infty$), γ the surface tension, ρ_v is the density of vapor, ρ_l the density of liquid, and r_1 and r_2 are the principal radii of the curved surface. The Gibbs-Thomson effect can be extended to solid-state nanoparticles to describe the surface concentration C of the particle atoms by^[R19]:

$$C = C_0 \exp\left(\frac{2\gamma\Omega}{kTr}\right), \quad (\text{R2})$$

where r is the particle radius, C_0 is the surface concentration of atoms in an infinitely large particle, Ω is the atomic volume, k is the Boltzmann constant and T is the temperature. For instance, Eq. (R2) can be used to describe Ostwald ripening of nanoparticles in the solid state.^[R20,R21]

Note that Eq. (R1) and Eq. (R2) are somehow equivalent, as they both predict that larger particles grow at the expense of smaller ones. The reason is that the surface of smaller particle has larger pressure (Eq. (R1)) or higher concentration (Eq. (R2)) of the particle atoms. However, neither Eq. (R1) nor Eq. (R2) can be used to directly predict the surface diffusion phenomenon and the evolution of the filament shape, which is instead obtained by Eqs. (1) and (2) in our numerical model. Yet, a key term in Eqs. (1) and (2) is the sum of inverse principal radii, similar to Eq. (R1). We thus conclude that, although our model has its roots in the Gibbs-Thomson effect, it stands as an original model, for both the formalism and the application. In other words, we are not reinventing the physics, but we are providing a new tool to physically describe an important phenomenon in RRAM technology, that is the universal volatile behavior.

We agree with the Referee that the key element in our theory, which differentiates our model from the Ostwald ripening, is that ‘in matrices, the surface (or should it be interface?) diffusion processes should change’. We rephrase this concept by saying that, in our system made of Ag or Cu filament within silk or SiO₂, the main physical process driving the filament shape evolution and controlling the device lifetime is the surface diffusion, rather than out-diffusion as in Ostwald ripening. This is because atomic movement at the surface (or interface) is easier than movement of atoms across the surrounding host material. While out-diffusion is essential to describe Ostwald ripening of separate particles^[R20,R21], surface diffusion is the most appropriate mechanism to minimize the surface-volume ratio within a single filament, or when particles are touching one to each other^[R22,R23]. The

Ostwald ripening has been proposed to control the evolution of the particles obtained by filament fragmentation [R26]. According to our model, surface diffusion might control the initial stages of the filament disconnection, which also dictates the filament lifetime according to the Herring's law. Ostwald ripening instead may be responsible for the post-lifetime evolution of the filament particles. In any case, the driving force for the surface diffusion and Ostwald ripening can be traced back to the Gibbs-Thomson effect.

We added one paragraph in the discussion part of the main manuscript (third paragraph of the section, page 8) to summarize the above discussion and a new section (Part XI) in the Supplementary Information for detailed explanation. The following references were also added in the manuscript to support the discussion.

[R18] Thomson, W. LX. On the equilibrium of vapour at a curved surface of liquid. *London, Edinburgh, Dublin Philos. Mag. J. Sci.* **42**, 448–452 (1871).

[R19] Simonsen, S. B. *et al.* Ostwald ripening in a Pt/SiO₂ model catalyst studied by in situ TEM. *J. Catal.* **281**, 147–155 (2011).

[R20] Simo, A. *et al.* Formation mechanism of silver nanoparticles stabilized in glassy matrices. *J. Am. Chem. Soc.* **134**, 18824–18833 (2012).

[R21] Simonsen, S. B. *et al.* Direct Observations of Oxygen-induced Platinum Nanoparticle Ripening Studied by In Situ TEM. *J. Am. Chem. Soc.* **132**, 7968–7975 (2010).

[R22] Kuczynski, G. C. Self-diffusion in sintering of metallic particles. *JOM* **1**, 169–178 (1949).

[R23] Mullins, W. W. Theory of thermal grooving. *J. Appl. Phys.* **28**, 333–339 (1957).

- *My second point is: I am assuming that the devices can switch multiple times. From the learnings of this work, and knowing that the devices “die” most often in the “ON” state, can one find a compromise between endurance and retention?*

Reply: Indeed, our devices can switch multiple times. It is not accurate to say that our devices ‘die’ in the ON state, rather they might remain in the ON state after switching at a high compliance current, which corresponds to the nonvolatile switching mode (Fig. 1d). The device can still be recovered to the off state by a negative voltage sweep. While the retention strongly depends on the type of switching, i.e., either volatile or nonvolatile, the endurance does not. There are reports in the literature where this type of device shows a high endurance for both volatile switching and nonvolatile switching. For instance, volatile RRAM devices made of Pd/Ag/HfO_x/Ag/Pd showed an endurance of 10⁸ [R24], AgNW/silk/AgNW of 10⁶ [R25], Pt/SiO_xN_y:Ag/Pt of 10⁶ [R26], Cu/HfO₂:Cu/Pt of 10¹⁰ [R27] while nonvolatile RRAM devices made of Cu/TaO_x/Pt showed an endurance of 10⁹ [R28], CuTe_x/Al₂O₃/TiN of 10⁶ [R29]. Since the same device can be operated in volatile or nonvolatile mode just by changing the compliance current I_C, one may wonder whether the device endurance is affected by I_C. Although there are currently no data comparing endurance on the same device operated in volatile mode (low I_C) or nonvolatile mode (high I_C), previous data on nonvolatile RRAM devices with HfO_x dielectric indicate no dependence of endurance on I_C [R30]. This was explained by the fact that a larger I_C is accompanied by a larger filament area, while the current density, which controls the local Joule heating and the associated device/material degradation, remains constant. We thus do not expect any dependence of endurance on the

volatile/nonvolatile switching mode, or, in other words, we do not expect a tradeoff between endurance and retention.

We added a new section (Part XII) in the Supplementary Information to incorporate the above discussion.

[R24] Midya, R. et al. Anatomy of Ag/Hafnia-based selectors with 10^{10} nonlinearity. *Adv. Mater.* **29**, 1604457 (2017).

[R25] Wang, M. et al. Enhancing the Matrix Addressing of Flexible Sensory Arrays by a Highly Nonlinear Threshold Switch. *Adv. Mater.* **30**, 1802516 (2018).

[R26] Wang, Z. et al. Memristors with diffusive dynamics as synaptic emulators for neuromorphic computing. *Nat. Mater.* **16**, 101–108 (2017).

[R27] Luo, Q. et al. Cu BEOL compatible selector with high selectivity ($> 10^7$), extremely low off-current (\sim pA) and high endurance ($> 10^{10}$). in 2015 IEEE International Electron Devices Meeting (IEDM) 10.4.1-10.4.4 (IEEE, 2015). doi:10.1109/IEDM.2015.7409669

[R28] Lv, H. et al. Evolution of conductive filament and its impact on reliability issues in oxide-electrolyte based resistive random access memory. *Sci. Rep.* **5**, 7764 (2015).

[R29] Robayo, D. A. et al. Statistical analysis of CBRAM endurance. in 2018 International Symposium on VLSI Technology, Systems and Application (VLSI-TSA) 1, 1–2 (IEEE, 2018).

[R30] Balatti, S. et al. Voltage-Controlled Cycling Endurance of HfO_x-Based Resistive-Switching Memory. *IEEE Trans. Electron Devices* **62**, 3365–3372 (2015).

- *In summary I find the paper appropriate for Nature Comms, however to have a more general impact, the Gibbs-Thomson phenomenon should be discussed/ruled out/accommodated so the reader can connect different fields and gather a more broad perspective. From a more area specific impact, I would like to see some discussion on retention x endurance. Could this work shed some light into this aspect? Are those two issues inextricably connected? After comments from the authors addressing these two points I find the paper suitable for publication.*

[1] see, for example: [dx.doi.org/10.1021/ja309034n](https://doi.org/10.1021/ja309034n), Simo et al., *J. Am. Chem. Soc.* 2012, 134, 18824–18833

Reply: We thank the Referee for the helpful comments again. The two main issues raised by the Referee (relevance of the Gibbs-Thomson effect and retention vs. endurance) were addressed in either the revised manuscript or the Supplementary Information. We made our best efforts to address the Referee's concerns and to solve the weak points raised by the Referee. We believe the manuscript has been significantly improved by these revisions, and we hope to find a favorable opinion by the Referee.

REVIEWERS' COMMENTS:

Reviewer #1 (Remarks to the Author):

The revised manuscript has addressed my previous questions/comments with further results and discussions. The work is now consistent and in good shape and should be published in Nature Communications.

Reviewer #2 (Remarks to the Author):

The authors have successfully addressed all my comments in the revised manuscript, which I believe is now suitable for publication in Nature Communications.

Reviewer #3 (Remarks to the Author):

The authors have complied with my suggestions, and I was able to gain a significantly deeper understanding on the matter and the work by itself. As such, I find it appropriate for publication.

Reply to the comments of Referee #1:

- *The revised manuscript has addressed my previous questions/comments with further results and discussions. The work is now consistent and in good shape and should be published in Nature Communications.*

Reply: We appreciate the authors positive comment. Thank you very much for your time and inspiring comments which allowed us to deeply improved our work.

Reply to the comments of Referee #2:

- *The authors have successfully addressed all my comments in the revised manuscript, which I believe is now suitable for publication in Nature Communications.*

Reply: We appreciate the authors positive comment. Thank you very much for your time and inspiring comments which allowed us to deeply improved our work.

Reply to the comments of Referee #2:

- *The authors have complied with my suggestions, and I was able to gain a significantly deeper understanding on the matter and the work by itself. As such, I find it appropriate for publication.*

Reply: We appreciate the authors positive comment. Thank you very much for your time and inspiring comments which allowed us to deeply improved our work.